# Performance of different nebulizers in clinical use for Pressurized Intraperitoneal Aerosol Chemotherapy (PIPAC)

Daniel Göhler[1,2], Kathrin Oelschlägel[1], Mehdi Ouaissi[3,4], Urs Giger-Pabst[3,5]*

**1** Topas GmbH, Dresden, Germany, **2** Research Group Mechanical Process Engineering, Institute of Process Engineering and Environmental Technology, Technische Universität Dresden, Dresden, Germany, **3** EA4245 Transplantation, Immunology, Inflammation, Université de Tours, Tours, France, **4** Department of Digestive, Oncological, Endocrine, Hepato-Biliary, Pancreatic and Liver Transplant Surgery, University Hospital of Tours, Tours, France, **5** Fliedner Fachhochschule, University of Applied Sciences Düsseldorf, Düsseldorf, Germany

* ursgiger@gmx.net

**Data Availability Statement:** All relevant data are within the paper.

**Funding:** The authors received no specific funding for this work.

## Abstract

### Objective

Technical ex-vivo comparison of commercial nebulizer nozzles used for Pressurized Intraperitoneal Aerosol Chemotherapy (PIPAC).

### Methods

The performance of four different commercial nebulizer nozzles (Nebulizer; HurriChem™; MCR-4 TOPOL®; QuattroJet) was analysed concerning: i) technical design and principle of operation, ii) operational pressure as function of the liquid flow rate, iii) droplet size distribution via laser diffraction spectrometry, iv) spray cone angle, spray cone form as well as horizontal drug deposition by image-metric analyses and v) chemical resistance via exposing to a cytostatic solution and chemical composition by means of spark optical emission spectral analysis.

### Results

The Nebulizer shows quasi an identical technical design and thus also a similar performance (e.g., mass median droplet size of 29 μm) as the original PIPAC nozzles (MIP/ CapnoPen). All other nozzles show more or less a performance deviation to the original PIPAC nozzles. The HurriChem™ has a similar design and principle of operation as the Nebulizer, but provides a finer aerosol (22 μm). The principle of operation of MCR-4 TOPOL® and QuattroJet differ significantly from that of the original PIPAC nozzle technology. The MCR-4 TOPOL® offers a hollow spray cone with significantly larger droplets (50 μm) than the original PIPAC nozzles. The QuattroJet generates an aerosol (22 μm) similar to that of the HurriChem™ but with improved spatial drug distribution.

**Competing interests:** The authors have declared that no competing interests exist.

## Conclusion

The availability of new PIPAC nozzles is encouraging but can also have a negative impact if their performance and efficacy is unknown. It is recommended that PIPAC nozzles that deviate from the current standard should be subject to bioequivalence testing and implementation in accordance with the IDEAL-D framework prior to routine clinical use.

## 1 Introduction

More than a decade ago, Pressurized Intraperitoneal Aerosol Chemotherapy (PIPAC) was introduced clinically as a new approach to deliver intraperitoneal chemotherapy to patients suffering from end-stage peritoneal surface malignancies. Using a high-pressure injector connected to a specially designed PIPAC nozzle, liquid chemotherapeutic drugs are aerosolised during laparoscopic surgery within the capnoperitoneum. This approach is expected to have a better spatial distribution pattern, higher depth of penetration, and increased drug concentration in the tissue than conventional liquid intraperitoneal chemotherapy [1, 2]. Clinical data from phase I/II and larger mono- and multicentre case series regarding safety, feasibility, and oncologic efficacy are encouraging. While the therapeutic role of PIPAC is still unclear [3], prospective randomized PIPAC trials are underway and their results are eagerly awaited [4, 5].

For over a decade, only the original PIPAC nozzle was available for clinical use, with more than 18'000 documented clinical applications worldwide were projected by the end of 2022 [3]. The original nozzle which was used for off-label PIPAC is a CE certified class IIa device to aerosolize aqueous drug solutions. The nebulizer produces a polydisperse, bimodal aerosol with a volume-weighted median droplet diameter of 25 μm with a full spray cone of approximately 70˚. Ex-vivo gravimetric studies demonstrated an inhomogeneous spatial distribution pattern with more than 86.0 vol.-% of the administered chemotherapy deposited by inertial impaction directly below the nozzle outlet on a circular surface with a diameter of 15 cm. Calculations reveal that under the given flow conditions the droplet size for a homogenous intra-abdominal drug distribution during PIPAC therapy should be approximately 1.2 μm. However, 97.5 vol.-% of the aerosolized liquid is composed of droplets with a diameter of $> 3$ μm [6]. To ensure comparability of the outcome data, much efforts was spent to standardize PIPAC therapy worldwide [7, 8]. But more recently, new nebulizer devices are also in clinical use. While the technical and clinical performance data of the original PIPAC nozzle has been extensively studied in pre- and clinical settings [6, 9], no or only very limited comparative data are available for the new PIPAC nozzles. Oncological surgeons around the world are now faced with the question of whether these newer nozzles are equivalent to the original nozzle technology or perhaps even have technical/functional advantages with a potentially better oncological outcome.

Based on the methodological findings on the technical characterization of the original PIPAC nozzle [6], this study aims to comparatively characterize the technical design, granulometric performance and spray characteristics (spray angle/cone) of all until the beginning of 2023 existing/known clinically operated nozzles for PIPAC. In addition, the chemo-resistance and metallurgical properties of the different nozzles were investigated by an independent, government-accredited testing agency.

## 2 Materials and methods

Excepting the investigation concerning chemical resistance and composition, all analyses took place in a non-technical-ventilated room (volume = 120.0 m$^3$) at a temperature of $\vartheta$ = (20.3 ± 1.3)°C, a relative humidity of $\varphi$ = (38.2 ± 3.7) % and a pressure of p = (101.6 ± 3.3) kPa.

### 2.1 Examined PIPAC nozzles

Four commercial single-substance PIPAC nozzles for intraperitoneal drug aerosolization were examined, i.e.,

- Nebulizer, Model 770–12, REGER Medizintechnik, Villingendorf, Germany (A),

- HurriChem™, ThermaSolutions, White Bear Lake, MN, United States of America (B),

- MCR-4 TOPOL®, SKALA-Medica, Soběslav, Czech Republic (C),

- QuattroJet, Model 770–14, REGER Medizintechnik, Villingendorf, Germany (D).

After the experiments, all nozzles were cut-open longitudinally in the middle in a 180° angle by means of a computerized numerical control milling machine to study their principles of operation. In addition, also the dimensions of the nozzle outlet orifices were examined by light microscopy (SMZ1500, Nikon, Tokyo, Japan).

### 2.2 Barometric characterisation of operational pressure as function of liquid flow rate

To characterise the operational pressure over the volumetric liquid flow rate, the nozzles were connected via high-pressure hose lines with a high-pressure injector (ACCUTRON® HP-D, MEDTRON AG, Saarbrücken, Germany) to push the test liquid (Glucosterile 5%, Fresenius Kabi GmbH, Germany) through the nozzles. The operational pressure induced by the liquid flow rate was determined by means of a glycerine-filled bourdon gauge (MA7U-25, JRA Maschinenteile und Geräte GmbH, Reichenbach, Germany), which was implemented in the high-pressure line. For the analyses, the volumetric liquid flow rate was increased stepwise either by 0.1 ml/s (for nozzles A, B and D) or by 0.2 ml/s (for nozzle C) until the maximum permitted pressure of 21 bar of the high-pressure injector was reached. For nozzle D, only the axial nozzle was tested—the horizontal nozzles were sealed watertight. Analogous to [6], measurement values were taken at steady state conditions of the aerosolization process and all analyses were repeated three times.

### 2.3 Granulometric characterisation of droplet size distributions

The droplet size distributions of the aerosols generated from the test liquid (Glucosterile 5%, Fresenius Kabi GmbH, Germany) were characterised by laser diffraction spectrometry (PW180-C spray particle size analyser, Jinan K-Ring Technology Co., Ltd, Shandong, China) over a size range of 0.57–780 μm. The outlets of the PIPAC nozzles were arranged via a tripod in a distance of 5 mm perpendicular to the centre of the free-accessible red laser beam. To characterise the aerosolization performance, all analyses were performed contemporaneous with the barometric characterisation of the operational pressure for various liquid flow rates. Analogous to [6], measurement values were taken at steady state conditions of the aerosolization process and all analyses were repeated three times. Due to geometric limitations of the operated laser diffraction spectrometer, only the aerosol that leaves the axial orifice of nozzle D could be analysed, while the lateral orifices had to be sealed.

## 2.4 Image-metric characterisation of spray cone angles, form and horizontal drug deposition areas

The spray cone angles, the form of the spray cones and the horizontal drug deposition area were characterised with different test liquids at nozzle-specific operation conditions as recommended by the manufacturers, i.e., at a volumetric liquid flow rate of 0.5 ml/s for the nozzles A, B, at 2.0 ml/s for nozzle C and 1.5 ml/s for nozzle D. The former two characteristics were evaluated on the base of a 5 wt.-% aqueous glucose solution (Glucosterile 5%, Fresenius Kabi GmbH, Germany), while the latter characteristic was assessed by operating the nozzles with undiluted royal blue ink (Pelikan Tinte 4001®, Hannover, Germany).

For the spray cone angle analyses, the nozzles were fixed on a tripod and vertically aligned. Photographic images were taken with a camera that was perpendicular positioned to the nozzle direction. The images were in-silico processed by overlaying with a digital 360˚ full-circle protractor for determining the spray cone angles.

The form of the spray cones was visualized by means of a line laser (GCL 2–15, Robert Bosch Power Tools GmbH, Leinfelden-Echterdingen, Germany) positioned in distance of 60 mm from the nozzle orifice at right angle into the spray cone. Fully evaluated spray cone forms were finally documented photographically.

The horizontal drug deposition on a level-aligned blotting paper was examined by operating the vertically aligned nozzles with a distance of 60 mm between the blotting paper and the nozzle orifice. The blotting paper was exposed for 3 s to the fully-developed spray jet. To achieve this, a mechanical diaphragm was placed in front of the spray jet. The diaphragm was opened automatically within 0.1 s, when the aerosol jet showed steady state nebulisation condition.

## 2.5 Assessing of chemical resistance and chemical composition

To assess the chemical resistance of the nozzle material against chemotherapeutic drugs, the nozzles were at first exposed to a cytostatic solution for 12 hours and afterwards stored in the dark at room temperature for 12 days within petri dishes. The chosen cytostatic solution was prepared in accordance to the mixture of high pressure/high dose PIPAC (HP/HD-PIPAC) [10], i.e., 6 mg of doxorubicin (Accord 2 mg/ml, Accord Healthcare GmbH, Munich, Germany) was admixed with 50 ml of a 0.9 wt.-% aqueous sodium chloride solution (Ecolav® 100, B. Braun, Melsungen, Germany). Finally, the nozzles were milled open in a laminar flow workbench and macroscopic changes were documented photographically.

Moreover, the chemical composition of the nozzles pipes was characterised for the elements C, Si, Mn, P, S, Cr, Ni, Mo, Cu, W and N by means of spark optical emission spectral analysis (SPECTROMAXx, SPECTRO Analytical Instruments GmbH, Kleve, Germany) via an accredited laboratory (WS Material Service GmbH, Essen, Germany).

## 3 Results

### 3.1 Technical design and principle of operation

The 90˚ sectional views of the head regions in Fig 1 show technical destails of the examined nozzles.

Externally, all nozzles consist of a stainless steel shaft (S) with a more or less pronounced nozzle head (H) on the lower part and a Luer lock thread on the upper part (not shown in Fig 1). The Luer lock threads serve for the connection of the nozzles with high-pressure injectors via high-pressure hose lines. Internally, the nozzles show partly considerable differences.

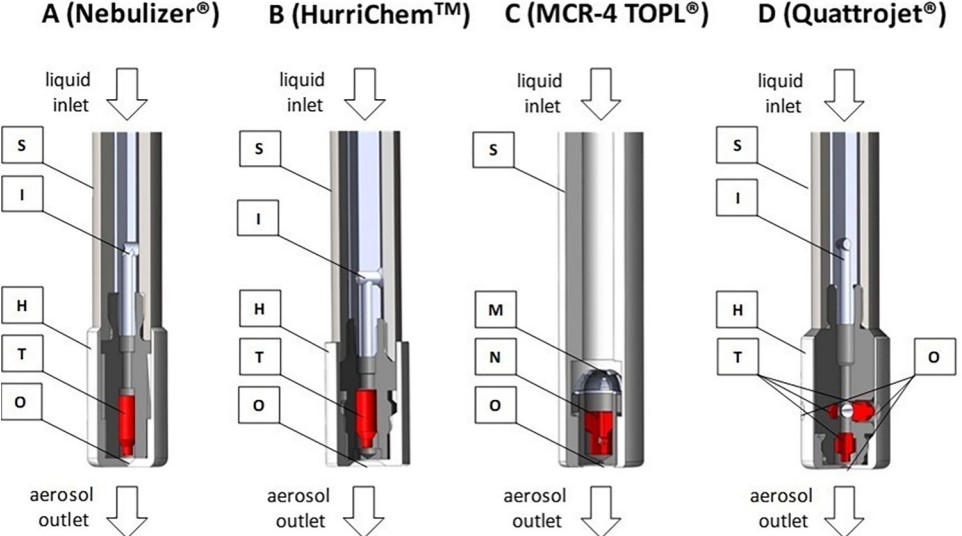

**Fig 1. 90˚ sectional views of the head regions of the nozzles.** O = outlet orifice; H = nozzle head; I = bar inlay with distal transverse borehole; M = double metal grid; N = fixed needle; S = shaft; T = twist body.

Interestingly, nozzle A and B are quasi identical in construction and their principle of operation, while the nozzles C and D differ significantly from them and from each other.

In the case of the nozzles A, B and D, the liquid drug is supplied internally from the Luer lock connector to the nozzle head via an annular gap between the outer shaft (S) and a bar inlay with distal transverse borehole (I). In contrast, the internal liquid drug supply of nozzle C occurs directly via the hollow cavity of the shaft (S). Moreover, nozzle C is equipped with a double metal grid (M) with two different mesh sizes that serve as particle filter.

While the nozzles A and B contain one twist body (T), nozzle D is equipped with four identical orifices and twist bodies (i.e., with one axial and three lateral twist bodies in 120˚ arrangement) to improve the spatial drug distribution within the abdominal cavity. The twist bodies (T) of the nozzles A, B and D contain longitudinally superficially milled grooves at 180˚ intervals. As the liquid drug flow rate passes along the twist bodies (T) they were set into rotation that improves the aerosolisation prior leaving the nozzle via the outlet orifice (O). In the case of nozzle C, the twist body is replaced by an fixed metal needle (N). This needle contains also laterally located, spirally milled axial grooves that induce a whirlwind effect for aerosolisation when passed by the liquid flow before leaving the nozzle via the outlet orifice (O).

Light microscopic images of the oulet orifices (O) with determined orifice diameters of the examined nozzles are shown in Fig 2.

## 3.2 Operational parameters based on barometric and granulometric analyses

Fig 3A depicts at first the determined operational pressure over the liquid flow rate of the examined nozzles, while in Fig 3B the mass median diameter of the of the droplet size distribution over the operational pressure is given. To avoid artefacts due clouding of the optics of the laser diffraction spectrometer, the lateral nozzles of nozzle D (QuattroJet) were taped off for the granulometric analyses and a flow rate of 0.5 ml/s was chosen (manufacturer-recommended flow rate of 1.5 ml/s). Note that the shown data are determined at steady state conditions of the aerosolization process analogous to [6].

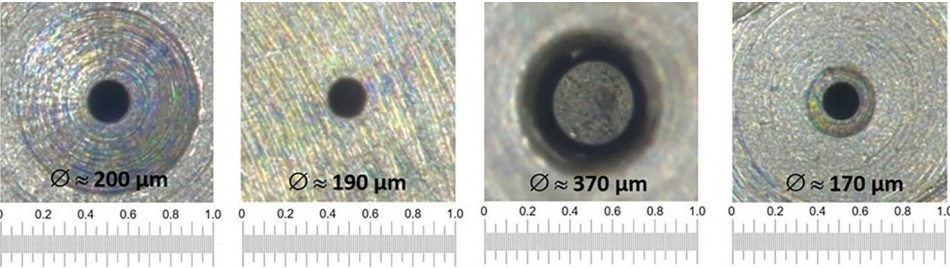

**Fig 2. Light microscopic images of the outlet orifices with determined orifice diameters of the examined nozzles; scaling in mm.**

It can be observed in Fig 3A that the determined operational pressure data for all examined nozzles fit well with the fluid dynamic theory, i.e., the dynamic pressure (or the dynamic pressure drop) of an incompressible fluid increase with the fluid velocity by the power of two. According to the equation of continuity, the fluid velocity of an incompressible fluid is in turn

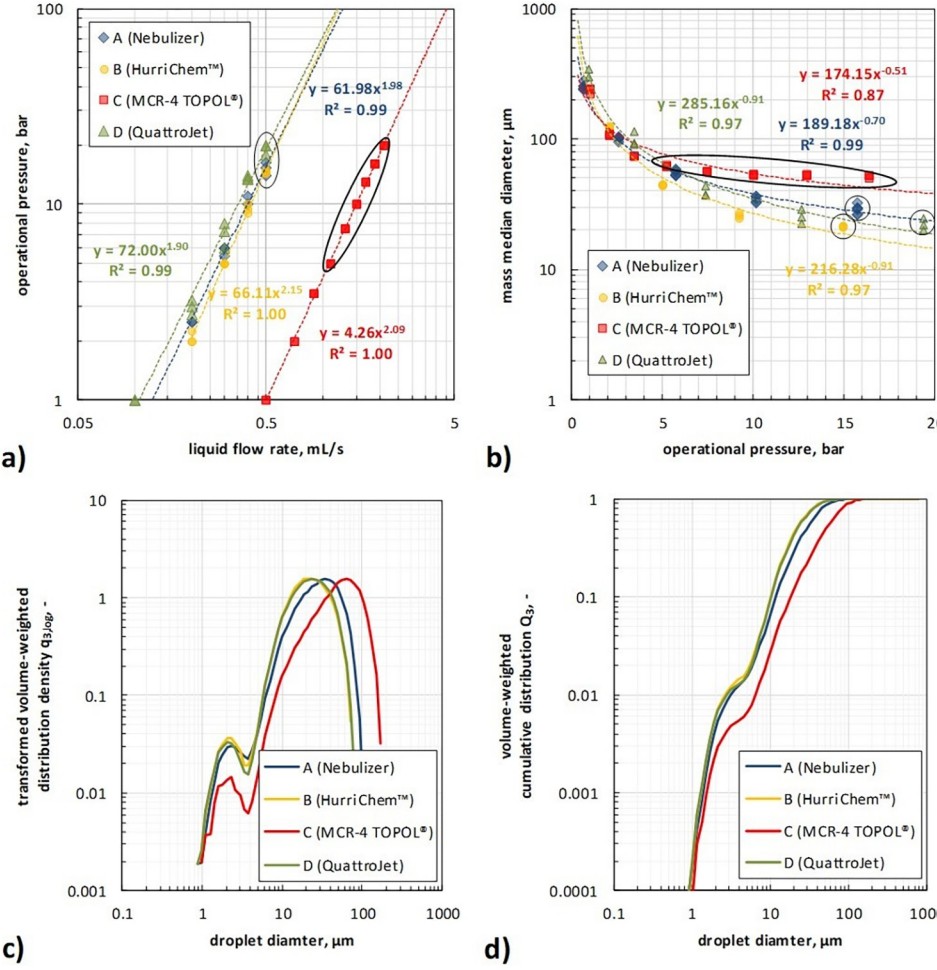

**Fig 3. Operational pressure as function of the liquid flow rate from barometric analyses (a), mass median diameter as function of the operational pressure from granulometric analyses (b) and volume-weighted distributions density (c) and cumulative distribution (d) of droplets at certain manufacturer-recommended operational condition; black cycles/ellipses indicate manufacturer-recommended operation condition.**

directly proportional to the volumetric liquid flow rate. The nozzles A, B and D show a similar performance regarding operational pressure and liquid flow rate, while nozzle C (MCR-4 TOPOL®) has a significantly lower pressure drop and thus a considerable higher volumetric liquid flow rate at a specific operational pressure.

Beside the whole operational spectrum, also the manufacturer-recommended operational conditions were separately examined, i.e., at a volumetric liquid flow rate of 0.5 ml/s for nozzle A (Nebulizer), nozzle B (HurriChem™), (1.3–2.0) ml/s for nozzle C (MCR-4 TOPOL®) and 1.5 ml/s for nozzle D (QuattroJet). Under these preconditions nozzle C showed with (18–26) s the shortest initiation time to reach the corresponding steady state pressure of (7.4–18.1) bar, followed by nozzle A with 52 s (15.7 bar) and nozzle D with 94 s (16.0 bar). Note that with taped-off lateral nozzles, nozzle D shows a higher operational pressure of 19.3 bar (as shown in Fig 3).

Fig 3B shows that the mass median diameter of the generated droplet aerosols depends for each nozzle significantly on the operational pressure. With increasing operational pressure, the mass median diameter decreases. For operational pressures of $\leq 4$ bar, quasi no significant difference between the different nozzles can be observed. This is attributed to a non-fully developed aerosolization of the supplied liquid. For operation pressures of $\geq 5$ bar stable aerosol generation is reached and differences between the nozzles can be observed. For operational pressures $\geq 5$ bar, nozzle C (MCR-4 TOPOL®) shows the coarsest mass median diameters, followed by nozzle A (Nebulizer). The finest mass median diameters were determined for nozzle B (HurriChem™) and D (QuattroJet). A statistical examination of the determined mass median diameters at manufacturer-recommended operation conditions (circled data in Fig 3B) shows on the basis of a Kruskal-Wallis H test [11] that the determined data for the analyzed nozzles are significantly different ($p_{A/B/C/D} = 0.000104$). Paired comparisons of the different nozzles using the Mann-Whitney test [12, 13] showed in the case of nozzle B (HurriChem™) and D (QuattroJet) quasi no significant difference ($p_{B/D} = 0.26$). In all other cases, determined mass median diameters of the different nozzles were found to be significant different to each other ($p_{A/B} = 0.0101$, $p_{A/C} = 0.0002$, $p_{A/D} = 0.0101$, $p_{B/C} = 0.0038$, $p_{C/D} = 0.0038$).

This ranking can also be deduced by the volume-weighted droplet size distributions of the aerosols as generated by the nozzles at the manufacturer-recommended operation conditions (Fig 3C and 3D). Moreover, it can be observed in Fig 3C and 3D, that each aerosol has a polydisperse and bimodal droplet size distribution.

## 3.3 Operational parameters based on image-metric analyses

Fig 4 shows photographic images for the spray cone angle (upper panel), the spray cone form (mid panel) and the horizontal drug deposition area (lower panel) of each examined nozzle as determined at manufacturer-recommended operational conditions.

According to the upper panel of Fig 4, the widest single spray cone angle was determined with 79˚ for nozzle C (MCR-4 TOPOL®), followed with 72˚ for nozzle A (Nebulizer) and with 71˚ for nozzle B (HurriChem™). Nozzle D (QuattroJet) shows with 67˚ the smallest single spray cone angel, but it has to keep in mind that nozzle D contains in contrast to the other nozzles of four spray cones. Moreover, it can be observed from the middle panel of Fig 4 that nozzle A (Nebulizer), nozzle B (HurriChem™) and nozzle D (QuattroJet) generate a full spray cone, whereas nozzle C (MCR-4 TOPOL®) produces a hollow spray cone. The full spray cones of the nozzles A, B and C lead also to complete filled circular areas of horizontal drug deposition beneath the nozzles as shown in the lower panel of Fig 4. In the case of the nozzles A and B, a circular deposition area of approx. 38.5 cm$^2$ (outer diameter of approx. 7 cm) was

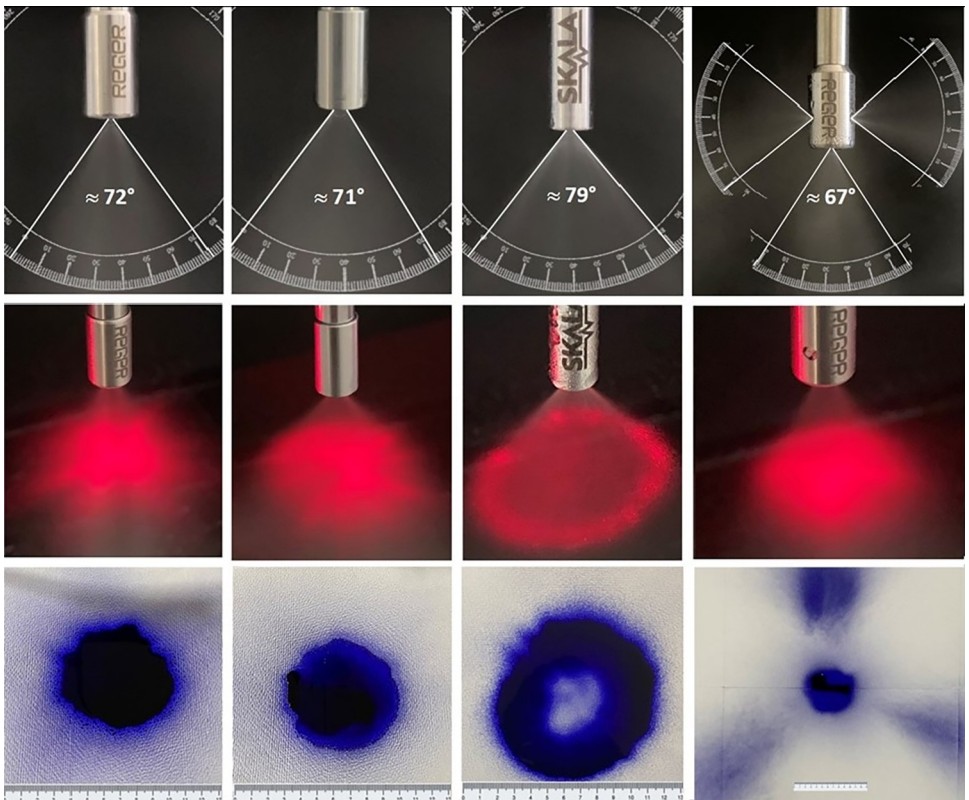

**Fig 4. Photographic images of spray cone angle (upper panel), of spray cone form (middle panel) and horizontal drug deposition area (lower panel, scale in cm).**

determined. The lateral outlets of nozzle D showed beside the axial circle (outer diameter of approx. 7 cm) also 3 additional deposition areas of $(13 \times 20)$ cm that cumulates to an overall horizontal deposition area of approx. 679 cm$^2$.

## 3.4 Chemical resistance and chemical composition

Photographic images of the nozzle parts after the exposure to the cytostatic solution are shown in Fig 5. Fig 5 shows that in the case of nozzle C (MCR-4 TOPOL®) the exposure to the cytostatic solution led to the formation of iron oxide. These are particularly pronounced on the fine-mesh particle filter, the nozzle needle and the nozzle head housing. No changes were observed for the nozzles A, B and D either visually or by light microscopic analyses.

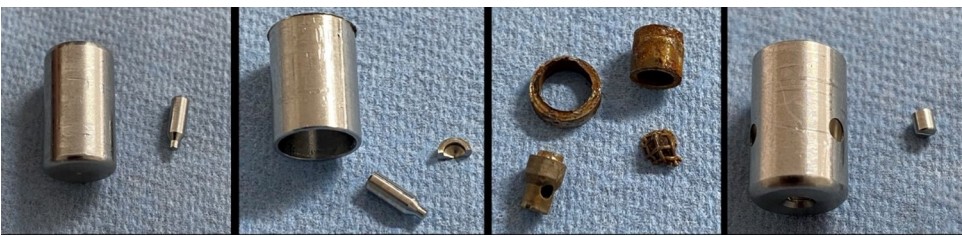

**Fig 5. Photographic images of the nozzle parts after exposure to the cytostatic solution.**

The nozzles A, B and D fulfil all requirements according to EN 10088–3:2014 [14] on the chemical composition of stainless steel 1.4301 that is typically used for surgical instruments. Beside a twelve times higher quantity of sulphur (0.012 wt.-% vs. 0.001wt.-%), also quantities of molybdenum (0.183 wt.-%), copper (0.220 wt.-%) and tungsten (0.134wt.-%) were identified by spark optical emission spectrometry for nozzle C (MCR-4 TOPOL®).

## 4 Discussion

Due to the current lack of knowledge, four clinically-operated nebulizing nozzles for PIPAC were comparatively tested regarding their performance. The most important determined technical characteristics of these nozzles are summarised in Table 1.

Nozzle A (Nebulizer) shows after an initiation time of 52 s an operational pressure of 15.7 bar at the manufacturer-recommended operational liquid flow rate of 0.5 ml/s. Thereby, a full spray jet cone (71°) composed of droplets with a mass median diameter of 29 μm is formed. The determined data of this study reveal that nozzle A is identical in design and performance to the primary for PIPAC developed predecessor, i.e., the microinjection pump MIP [6], which was also distributed under the tradename CapnoPen.

Nozzle B (HurriChem™) is another launched nebulizer for PIPAC. Examinations on design and principle of operation show a high similarity with nozzle A (Nebulizer) and thus also with the initial PIPAC nozzle technology. At the manufacturer-recommended operational liquid flow rate of 0.5 ml/s, nozzle B shows after an initiation time of 100 s an operational pressure of 14.9 bar. The mass median diameter of the droplets in the formed full spray jet cone (73°) was determined to be 21 μm.

Nozzle C (MCR-4 TOPOL®) differs in technical design, principle of operation, operational parameter and aerosol characteristics significantly from all other investigated nozzles. The operation of nozzle C is accompanied by the formation of a hollow spray cone jet (79°). At the manufacturer-recommended operational liquid flow rate range of (1.3–2.0) ml/s, operational pressures of (7.4–18.1) bar were reached within short initiation times of (18–26) s. The mass median droplet size decreases with increasing liquid flow rate, but was found to be in each case larger than 50 μm.

Nozzle D (QuattroJet) is a further PIPAC nebulizing nozzle that was introduced by the manufacturer of nozzle A. To optimize the spatial drug distribution pattern and higher intraabdominal aerosol particle concentration, the conventional axial nozzle is supplemented in nozzle D by three further nozzles, which are arranged lateral at the nozzle head with an angular distance of 120°. Nozzle D is based on the same technology as nozzle A. At the manu-facturer-recommended flow rate of 1.5 ml/s, nozzle D shows an operational pressure of 16.0

**Table 1.  Overview on technical and functional characteristics of the examined nozzles; * = manufacturer-recommended operational conditions.**

| parameter | unit | A | B | C | D |
|---|---|---|---|---|---|
| | | (Nebulizer) | (HurriChem™) | (MCR-4 TOPOL®) | (QuattroJet) |
| *liquid flow rate (*$Q_L$) | ml/s | 0.5 | 0.5 | 1.3–2.0 | 1.5 |
| operational pressure for *$Q_L$ | bar | 15.7 | 14.9 | 7.4–18.1 | 16.0 |
| pressure initiation time for *$Q_L$ | s | 52 | 100 | 18–26 | 94 |
| nozzle orifice diameter | μm | 200 | 190 | 370 | 170 |
| mass median diameter for 15 bar | μm | 28.95 | 20.99 | 52.17 | 24.18 |
| max. spray angle for *$Q_L$ | ° | ≈ 72 | ≈ 71 | ≈ 79 | ≈ 67 |
| number of nozzles | - | 1 × axial | 1 × axial | 1 × axial | 1 × axial, 3 × lateral |
| kind of spray cone | - | full cone | full cone | hollow cone | full cone |
| drug deposition area for *$Q_L$ | cm² | ≈ 38.5 | ≈ 38.5 | ≈ 66 | ≈ 679 |

bar after an initiation time of 92 s and provides four full spray cone jets (67°) composed of droplets with a mass median diameter of 21 μm.

Recently, a first attempt regarding recommendations on the minimum technical requirements on nozzles suitable for PIPAC treatment was published. A minimum requirement for the spray angle of at least 70° was defined [15] by implying that the spray cone angle corresponds to the achievable drug deposition area. But there are two limitations found in this study that contradict this requirement. The present study shows that the requirement is matched directly by nozzle A, B and C. But despite of a slightly lower spray cone angle of 67° than required, nozzle D consists of four spatially-displaced spray jets cumulating in a total spray angle of 268°. On the other hand, Nozzle C, unlike all other nozzles examined, had a hollow spray cone, resulting in a ring-shaped drug deposition area that was smaller than that of a full spray cone jet at the same spray cone angle. With regard on the nozzle performance, the drug deposition area seems to be an even better technical parameter than the spray cone angle.

The nozzles C (MCR-4 TOPOL®) and D (QuattroJet) investigated in this study, in contrast to A (Nebulizer) and B (HurriChemTM) and thus in contrast to the primary PIPAC nozzle technology, show significant differences in their operating principle and performance. Nozzle C offers the largest spray cone angle of all examined nozzles, but also a hollow spray cone. It is not evident if such a spray jet improves drug distribution and drug penetration, since so far as known there are no preclinical studies comparing a hollow with a full spray cone. Nozzle D provides multiple spray cones that can significantly improve the spatial drug distribution by reduction of high local deposition and thus high local tissue toxicity. Nonetheless, these potential benefits of multi-nozzle systems need to be confirmed by further research.

A worrying result of this study was, that nozzle C (MCR-4 TOPOL®) shows in contrast to the other nozzles a macroscopically visible formation of iron oxide after long-term exposure in a sodium chloride containing cytostatic solution. Based on spark optical emission spectrometry, it was found that this nozzle was made from a steel that, in addition to a 12-fold higher sulphur content (0.012 wt.-% vs. 0.001 wt.-%), also contained amounts of molybdenum (0.183 wt.-%), copper (0.220 wt.-%) and tungsten (0.134 wt.-%). Short-term exposure of the nebulizers to cytostatic solution shows no immediate corrosion, but a possible risk for the patient cannot be completely ruled out. This leads to the conclusion that manufacturer and regulatory authorities for medical devices accreditation must also critically assess the suitability of specific steel alloys for the administration of cytostatic drugs in their risk assessment.

Currently, there are only limited preclinical data to suggest that there is an optimal technique for the generation and delivery of PIPAC aerosols that could improve clinical outcome. However, it is clear that, contrary to claims made by one manufacturer [16], larger aerosol droplets injected into the peritoneal cavity at higher velocities with a hollow spray cone do not improve either the spatial distribution pattern [6, 17] or tissue penetration depth per se. Nebulizers differing from the present standard technology in its design, especially in its spraying characteristics, cannot automatically be considered equivalent by the clinical user. Therefore, before their broad clinical use, the individual innovation phases should be systematically tested, ideally following the recommendation of the IDEAL-D framework for the introduction of medical devices. IDEAL-D is an independent network of clinicians and scientists that aims to provide a framework for evaluating innovations in surgical techniques from conception, through technical development, research, evaluation and long-term follow-up. The central tenet of this framework is that development and evaluation can and should be done together in an orderly and logical manner that balances innovation and safety. The evaluation for interventional therapies in the IDEAL-D framework includes four stages comprising the preclinical phase (stage 0), first-in-human studies (stage 1), prospective developmental/exploratory studies (stage 2), large randomized controlled trials/or equivalent studies (stage 3) and long-term

surveillance studies and registries (stage 4) [18, 19]. While the original nozzle technology has completed phase I—IIb (IDEAL-D stage 2) [20] and phase III trials (IDEAL-D stage 3) are ongoing [4, 5], only limited phase I (IDEAL-D stage 1) clinical user data has been published for nozzle C (MCR-4 Topol®) [21]. Such data are lacking for nozzle B and D (IDEAL-D stage 0).

The extent to which the PIPAC nozzle designs and thus its' spray characteristics affect the spatial distribution, penetration depth and tissue concentration of drugs has hardly been investigated so far. Göhler et al. [22] compared the performance of the original PIPAC nozzle with that of a two-substance nozzle in a post-mortem pig model and found that due to the even more finer droplet size distribution of the latter one, the spatial drug distribution of $^{99m}$Tc pertechnetate as well as the penetration depth of doxorubicin becomes significantly more homogeneously. Mun et al. [23] presented a nozzle (DreamPen®) with a spray cone angle of 77° that rotates 30° around its vertical axis during operation within the abdominal cavity. The direct comparison with the original PIPAC nozzle in an in-vivo pig model showed an improved spatial drug distribution pattern of methylene blue and significantly higher penetration depths or tissue concentrations of doxorubicin in the peritoneum [24]. In a recent study, Braet et al. [25] administered nanoparticles in an ex-vivo model via an original PIPAC nozzle and a multidirectional nebulizer (Medspray®), each with and without additional electrostatic aerosol precipitation. It was found that the addition of electrostatic aerosol precipitation improves the spatial distribution of the nanoparticles and their penetration depth into the tissue, regardless of the nozzle used. Additionally, the operation of the multidirectional nozzle showed that the spatial distribution pattern of the nanoparticles was more homogeneous than for the conventional PIPAC nozzle. To our knowledge, neither the two-substance nozzle nor the DreamPen® or the Medspray® nebulizer is approved for clinical use. In summary, the mentioned preclinical studies show that there is large space for advanced technologies that can improve the efficacy of PIPAC. But the impact on oncologic outcome is still unclear and requires extensive preclinical and clinical research in the near future.

In the near future, clinical users should be able to rely on technical testing and reporting being based on scientific standards and generally applicable global standards, such as ISO standards. For PIPAC nebulizers, such standards should ideally be set by a panel of experts through consensus conferences. Moreover, nebulizers with significant technical and granulometric differences from the standard technology should undergo first ex- and in-vivo animal testing before their first clinical use [26–34]. Manufacturers should be obliged to have the bioequivalence of the cytostatic drugs administered independently certified in comparison to standard nebulizer systems, analogous to drugs and their generics. Relevant outcome measures are aerosol characteristics, spatial drug distribution, depth of penetration, tissue concentration and the peak concentration and the area under the curve describing the extent of peritoneal passage. The ratio between the individual properties of the generic nebulizer and the reference product would ideally be 1:1 in case of bioequivalence. As this is unlikely to be achieved, the US Food and Drug Administration (FDA), for example, requires the 90% confidence interval for drugs and their generics to be between 0.80 and 1.25 [35]. Similar to such FDA specifications, new PIPAC nozzle technologies could be tested comparatively in the future. Such preclinical testing, ideally using standardized models, could prevent the use of such devices from compromising clinical outcomes and/or harming healthcare professionals/patients. Finally, it would be helpful for the comparability of clinical results if the nebulizer type used in each case will be also recorded in the PIPAC database (https://isspp.org/professionals/pipac-database/).

It is pointed out that the following remarks and limitations have to be taken into account when evaluating the results of this study. First, one has to keep in mind that all barometric and granulometric analyses were performed on the basis of an aqueous solution of glucose (5 wt.-%). These data lose their validity, if liquids with deviating viscosity (such as drug carrier gels,

nanoparticles, antibody solutions, etc.,) are nebulized for research or clinical purposes. Second, only one specimen of each type of nozzle was examined at least three time, i.e., the impact of manufacturing tolerances on the performance of the nozzles is unknown. Third, due to geometric limitations of the operated laser diffraction spectrometer, only the aerosol that leaves the axial orifice of nozzle D (QuattroJet) could be analysed. Fourth, the operated laser diffraction spectrometer (PW180-C spray particle size analyser, Jinan K-Ring Technology Co., Ltd, Shandong, China) was successfully validated in side-analyses against a second laser diffraction spectrometer (HELOS/KR-H2487, Sympatec GmbH, Clausthal-Zellerfeld, Germany). Ultimately, when a nebulizing system for PIPAC is clinically used, the responsibility for compliance with national/supranational regulations for the use of medical devices lies with the user.

## 5 Conclusion

Four clinically-used nozzles to aerosolise chemotherapeutic drugs in the context of Pressurized Intraperitoneal Aerosol Chemotherapy (PIPAC), i.e., the Nebulizer, the HurriChem™, the MCR-4 TOPOL® and the QuattroJet were examined comparative to determine their performance.

It could be confirmed that the Nebulizer shows quasi an identical technical design and thus also a similar performance as the original PIPAC nozzles MIP/CapnoPen. The PIPAC nozzle HurriChem™ is based on a similar technical design as the Nebulizer nozzle, but provides a finer aerosol due to a smaller nozzle orifice opening. Both, the MCR-4 TOPOL® and the QuattroJet deviate in the principles of operations to that of the Nebulizer and thus to the original PIPAC technology. While the MCR-4 TOPOL® provides the coarsest aerosol of all examined nozzles, the QuattroJet delivers an aerosol similar to that of the HurriChem™. In contrast to the HurriChem™, the QuattroJet comes with the feature of four spray nozzles (one axial, three lateral) to improve the spatial drug distribution and a higher aerosol particle number concentration.

The availability of new PIPAC nozzles with special features is encouraging, but can also have a negative impact for establishment of the promising PIPAC approach for the treatment of peritoneal carcinomatosis, if their performance and efficacy is unknown. It is therefore recommended that nozzles for which the technical/granulometric characteristics differ from the current standard technology must be subjected to preclinical proof of equivalence in terms of spatial drug distribution, tissue penetration and concentration before routine clinical use. It is expected that representative in-silico models like the ones used in [36, 37] will be available in the near future. New nebulizers should be investigated and introduced for clinical use in accordance with the IDEAL-D framework.

## Acknowledgments

The authors thank Professor Marc Pocard, Hepato-Biliary-Pancreatic Gastrointestinal Surgery and Liver Transplantation, Pitié Salpêtrière Hospital, AP-HP, F-75013 Paris, France, for providing a HurriChem™ (ThermaSolutions, White Bear Lake, MN, USA) nozzle.

## Author Contributions

**Conceptualization:** Daniel Göhler, Mehdi Ouaissi, Urs Giger-Pabst.

**Data curation:** Daniel Göhler, Mehdi Ouaissi, Urs Giger-Pabst.

**Formal analysis:** Daniel Göhler, Kathrin Oelschlägel, Mehdi Ouaissi, Urs Giger-Pabst.

**Investigation:** Daniel Göhler, Mehdi Ouaissi, Urs Giger-Pabst.

**Methodology:** Daniel Göhler, Kathrin Oelschlägel, Mehdi Ouaissi, Urs Giger-Pabst.

**Project administration:** Kathrin Oelschlägel, Urs Giger-Pabst.

**Resources:** Daniel Göhler, Kathrin Oelschlägel, Mehdi Ouaissi, Urs Giger-Pabst.

**Software:** Daniel Göhler.

**Supervision:** Daniel Göhler, Kathrin Oelschlägel, Mehdi Ouaissi, Urs Giger-Pabst.

**Validation:** Daniel Göhler, Kathrin Oelschlägel, Mehdi Ouaissi, Urs Giger-Pabst.

**Visualization:** Daniel Göhler, Kathrin Oelschlägel, Mehdi Ouaissi, Urs Giger-Pabst.

**Writing – original draft:** Daniel Göhler, Kathrin Oelschlägel, Mehdi Ouaissi, Urs Giger-Pabst.

**Writing – review & editing:** Daniel Göhler, Kathrin Oelschlägel, Mehdi Ouaissi, Urs Giger-Pabst.

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
