## [Decision Letter · Decision Letter 0]

4 Sep 2023

PONE-D-23-07734

Performance of different nebulizers in clinical use for Pressurized Intraperitoneal Aerosol Chemotherapy (PIPAC)

PLOS ONE

Dear Dr. Giger-Pabst,

Thank you for submitting your manuscript to PLOS ONE. After careful consideration, we feel that it has merit but does not fully meet PLOS ONE’s publication criteria as it currently stands. Therefore, we invite you to submit a revised version of the manuscript that addresses the points raised during the review process.

We look forward to receiving your revised manuscript.

Kind regards,

Mohammad Mehdi Rashidi

Academic Editor

PLOS ONE

Journal Requirements:

Reviewers' comments:

Reviewer's Responses to Questions

**Comments to the Author**

1. Is the manuscript technically sound, and do the data support the conclusions?

Reviewer #1: Yes

Reviewer #2: Partly

Reviewer #3: Yes

2. Has the statistical analysis been performed appropriately and rigorously? 

Reviewer #1: Yes

Reviewer #2: N/A

Reviewer #3: Yes

3. Have the authors made all data underlying the findings in their manuscript fully available?

Reviewer #1: Yes

Reviewer #2: Yes

Reviewer #3: Yes

4. Is the manuscript presented in an intelligible fashion and written in standard English?

Reviewer #1: Yes

Reviewer #2: Yes

Reviewer #3: Yes

5. Review Comments to the Author

Reviewer #1: 1. Overview

This study aimed to compare the technical performance of four commercial nebulizer nozzles used for Pressurized Intraperitoneal Aerosol Chemotherapy (PIPAC). The analysis included technical design, operational pressure, droplet size distribution, spray cone angle and form, horizontal drug deposition, chemical resistance, and chemical composition. The Nebulizer showed a similar technical design and performance to the original PIPAC nozzles, while the other nozzles demonstrated deviation from the original nozzles. The study recommends that PIPAC nozzles that deviate from the current standard undergo bioequivalence testing and follow the IDEAL-D framework before clinical use.

2. Major Comments

The paper presents a meaningful and rigorous study of the four types of nebulizer nozzles. The experiment results and suggestions are important to guide new pre-clinical screening to determine its equivalency. In addition to the experiment and discussion in the paper, it would be helpful to include the information below.

The paper mentions multiple times that the aerosol generation principle is different between A/B, C, and D. It would be helpful to elaborate further on how the aerosol is generated in each device. A zoom-in schematic diagram of the liquid aerosolization in each nebulizer may further assist in the understanding of their performance difference.

The nebulizers are eventually used for PIPAC. It would be important to cite and summarize the drug delivery requirement and guidelines of the PIPAC. The discussion would be more meaningful in the context of its actual application environment.

In addition to suggesting equivalency studies, the result here could further guide the development of new nebulizer structures that combine the advantage and avoid the drawback of each design. At the end of the discussion, the author could provide such suggestions for new device development.

3. Minor Comments

Page 5 line 120, throughout the testing, the horizontal nozzles were mostly sealed and not tested. It would be helpful to include data where horizontal nozzles are opened while the axis nozzle is sealed. Combining the two experiments could provide a whole picture of the nozzle D performance.

Fig 4, it is apparent from the bottom panel that the density of aerosols has a special distribution. Further image analysis of the color intensity and integration of the intensity over the whole area could give a quantitative estimation of the drug dose. If possible, such integration over 3D space is even better.

Page 6 line 158, it is alarming that nozzle C forms rust over long exposure to the cytostatic solution. However, given their different standard operation conditions (see Page 6 line 138), nozzle C has the highest flow rate. To make a fair comparison assuming the same drug dose, nozzle C’s testing time or salt concentration might be scaled down compared to nozzle A/B/D.

Reviewer #2: This study aimed to compare four commercial nebulizer nozzles used for Pressurized Intraperitoneal Aerosol Chemotherapy (PIPAC). The Nebulizer showed similar performance to the original PIPAC nozzles, while the HurriChemTM offered a finer aerosol, MCR-4 TOPOL® had larger droplets, and QuattroJet improved spatial drug distribution. The authors recommend conducting bioequivalence testing and adhering to the IDEAL-D framework before implementing new PIPAC nozzles with different performances in routine clinical use.

Introduction: The introduction provides a good background on PIPAC, its clinical significance, and the need for standardized nozzles. However, it would be beneficial to include a clearer statement of the research objective or hypothesis to guide the readers better.

Materials and Methods: The methods section is quite detailed, which is good for reproducibility. However, it would be helpful to include information about the experimental setup, such as the type of equipment used, and some basic parameters like temperature and humidity during the experiments, as these factors can influence the aerosolization process.

The study design and methodology seem appropriate for the research objective. However, additional details on the experimental setup, such as the type of high-pressure injector used and any potential variations in environmental conditions (temperature, humidity), should be included to ensure reproducibility by other researchers.

While the results are presented in a clear manner, it would be beneficial to include statistical analyses to determine the significance of the differences observed between the nozzles, especially when comparing operational pressure and droplet size distribution.

The chemical resistance of the nozzles is an important factor, especially when considering exposure to cytostatic solutions. A more detailed discussion of the implications of iron oxide formation in nozzle C and its potential impact on nozzle performance or patient safety should be included.

Clinical Relevance: While the technical and operational comparisons are well-done, it would be valuable to discuss the potential clinical implications of the observed differences in nozzle performance. How might these differences affect drug delivery and patient outcomes during PIPAC procedures?

Limitations: Include a section on limitations to acknowledge any potential biases or shortcomings in the study. For instance, limitations related to the sample size, choice of test liquids, or specific measurement techniques should be discussed.

Since the angioinjector is not yet standardized, it is necessary to investigate the potential changes that may occur when using different types of devices. Therefore, a comprehensive evaluation of the effects of using alternative devices is required. This includes assessing any potential variations in operational parameters, droplet size distributions, spray cone angles, drug deposition areas, and chemical resistance compared to the angioinjector. Understanding these differences will be crucial for determining the clinical implications and safety of using different devices for Pressurized Intraperitoneal Aerosol Chemotherapy (PIPAC).

To establish a direct clinical relevance and connection, it is essential to consider parameters such as penetration depth and tissue concentration. However, it is evident that these factors may present limitations in this study. Since the angioinjector has not been standardized, variations in operational parameters and droplet characteristics among different devices could potentially impact penetration depth and drug tissue concentration. Thus, to fully understand the clinical implications and efficacy of using alternative devices for Pressurized Intraperitoneal Aerosol Chemotherapy (PIPAC), further research and validation studies are warranted. These investigations should focus on the actual clinical outcomes, including tumor response rates, patient outcomes, and adverse events, while considering the impact of different device designs and performance characteristics on drug delivery to the targeted tissues.

Addressing the regulation issues related to nozzles in different countries would indeed add more depth to the content. Currently, there might be varying regulatory frameworks and standards governing the approval and use of nebulizer nozzles for PIPAC in different countries. Understanding these variations is crucial as it may impact the availability, accessibility, and usage of different nozzle types globally. Additionally, these regulatory differences may also affect the acceptance and adoption of newer nozzle technologies in specific regions, making it necessary to navigate through regulatory hurdles before implementing them in routine clinical practice. Therefore, exploring the regulatory landscape in different countries will provide valuable insights into the challenges and opportunities surrounding the usage of various PIPAC nozzles worldwide.

Reviewer #3: The authors have tested clinically-operated nebulizing nozzles for PIPAC: HurriChemTM; MCR-4 TOPOL®; QuattroJet) for i) technical design and principle of operation, ii) operational pressure as a function of the liquid flow rate, iii) droplet distribution via laser diffraction spectrometry, iv) spray cone angle, spray cone form as well as horizontal drug deposition by image-metric analyses and v) chemical resistance via exposing to a cytostatic solution and chemical composition by means of spark optical emission spectral analysis.

The authors have done a detailed and sound study of PIPAC nozzles which can make it easier to choose the best nozzle for chemotherapy in the future. It can help manufacturers and clinicians both in making better choices regarding cancer-type-specific PIPAC nozzle devices.

Accept after minor revisions

The authors have done a good job and the subject of the work in itself is exploration worthy and holds promising applications. PIPAC devices have become important for chemotherapy. This device has shown promise in improving median survival, especially in the case of ovarian and colorectal metastasis (https://www.ncbi.nlm.nih.gov/pmc/articles/PMC8683254).

A few suggestions to make the manuscript more meaningful are mentioned below:

1. What is the basis for choosing four commercial nebulizers from all the available options? What are the other available nozzles for PIPAC? Can we include a table of such nozzles and explain the choices made in the manuscript?

2. Lin 94: there is a question mark, please remove it. Also, write briefly about the “original nozzle technology in the same paragraph.

3. Line 72: Please give an introductory detail of the IDEAL-D framework somewhere in the manuscript to make it easy for the readers. A reference has already been given which is appreciated.

4. How is the mass median diameter generated from the nozzles related to the efficacy of treatment? Are we expecting a similar kind of droplet distribution when popular chemotherapeutic drugs like cisplatin, etc will be used? How will the aerosolization and droplet distribution change with changing molecular weight of the drugs? Can we think of some in vivo studies on cancer tissue slices or a plate-based assay to test the efficacy of cell survival? How are the functional parameters explored related to the killing the cancer cells, are there any related studies? Please include a little bit of such discussion in the manuscript to make it easy to be understood by people of the interdisciplinary fields.

6. PLOS authors have the option to publish the peer review history of their article (what does this mean?). If published, this will include your full peer review and any attached files.

Reviewer #1: No

Reviewer #2: No

Reviewer #3: No

---

## [Author Response · Author response to Decision Letter 0]

20 Oct 2023

All responses on revierwer and editor comments can be found in the uploaded response letter.

---

## [Decision Letter · Decision Letter 1]

11 Jan 2024

PONE-D-23-07734R1Performance of different nebulizers in clinical use for Pressurized Intraperitoneal Aerosol Chemotherapy (PIPAC)PLOS ONE

Dear Dr. Giger-Pabst,

Thank you for submitting your manuscript to PLOS ONE. After careful consideration, we feel that it has merit but does not fully meet PLOS ONE’s publication criteria as it currently stands. Therefore, we invite you to submit a revised version of the manuscript that addresses the points raised during the review process.

We look forward to receiving your revised manuscript.

Kind regards,

Mohammad Mehdi Rashidi

Academic Editor

PLOS ONE

Journal Requirements:

Reviewers' comments:

Reviewer's Responses to Questions

**Comments to the Author**

1. If the authors have adequately addressed your comments raised in a previous round of review and you feel that this manuscript is now acceptable for publication, you may indicate that here to bypass the “Comments to the Author” section, enter your conflict of interest statement in the “Confidential to Editor” section, and submit your "Accept" recommendation.

Reviewer #1: All comments have been addressed

2. Is the manuscript technically sound, and do the data support the conclusions?

Reviewer #1: Yes

3. Has the statistical analysis been performed appropriately and rigorously? 

Reviewer #1: Yes

4. Have the authors made all data underlying the findings in their manuscript fully available?

Reviewer #1: Yes

5. Is the manuscript presented in an intelligible fashion and written in standard English?

Reviewer #1: Yes

6. Review Comments to the Author

Reviewer #1: Dear Author, thank you for addressing my comments with a revision. I could not find your point-to-point response via the online portal, but I was able to see the changes you made in the revision. I have no further comments. However, please DO correct lines 264-286 "Error! Reference source not found," prior to publication.

7. PLOS authors have the option to publish the peer review history of their article (what does this mean?). If published, this will include your full peer review and any attached files.

Reviewer #1: No

---

## [Author Response · Author response to Decision Letter 1]

14 Jan 2024

please see rebuttal letter or respond to reviewers

---

## [Decision Letter · Decision Letter 2]

26 Feb 2024

Performance of different nebulizers in clinical use for Pressurized Intraperitoneal Aerosol Chemotherapy (PIPAC)

PONE-D-23-07734R2

Dear Dr. Giger-Pabst,

We’re pleased to inform you that your manuscript has been judged scientifically suitable for publication and will be formally accepted for publication once it meets all outstanding technical requirements.

Kind regards,

Fabrizio D'Acapito, Ph.D,M.D.

Academic Editor

PLOS ONE

Additional Editor Comments (optional):

I congratulate the authors for their work. I believe that all the annotations of previous reviewers have been adequately addressed.

The topic is challenging particularly for surgeons who are the main users of PIPAC. The images and tables associated with the text are effective in aiding the understanding of engineering notions.

I believe it is important to have provided evaluation of the tool that plays a key role in the efficacy of a therapeutic method that may change the use of intraperitoneal chemotherapy in the near future.

Reviewers' comments:

Reviewer's Responses to Questions

**Comments to the Author**

1. If the authors have adequately addressed your comments raised in a previous round of review and you feel that this manuscript is now acceptable for publication, you may indicate that here to bypass the “Comments to the Author” section, enter your conflict of interest statement in the “Confidential to Editor” section, and submit your "Accept" recommendation.

Reviewer #1: All comments have been addressed

2. Is the manuscript technically sound, and do the data support the conclusions?

Reviewer #1: Yes

3. Has the statistical analysis been performed appropriately and rigorously? 

Reviewer #1: Yes

4. Have the authors made all data underlying the findings in their manuscript fully available?

Reviewer #1: Yes

5. Is the manuscript presented in an intelligible fashion and written in standard English?

Reviewer #1: Yes

6. Review Comments to the Author

Reviewer #1: (No Response)

7. PLOS authors have the option to publish the peer review history of their article (what does this mean?). If published, this will include your full peer review and any attached files.

Reviewer #1: No
